# A Dual-Acting Nitric Oxide Donor and Phosphodiesterase 5 Inhibitor Activates Autophagy in Primary Skin Fibroblasts

**DOI:** 10.3390/ijms23126860

**Published:** 2022-06-20

**Authors:** Esther Martínez-Martínez, Paola Atzei, Christine Vionnet, Carole Roubaty, Stephanie Kaeser-Pebernard, Reto Naef, Jörn Dengjel

**Affiliations:** 1Department of Biology, University of Fribourg, 1700 Fribourg, Switzerland; emartinez@cipf.es (E.M.-M.); christine.vionnet@unifr.ch (C.V.); carole.roubaty@unifr.ch (C.R.); stephanie.kaeser-pebernard@unifr.ch (S.K.-P.); 2Topadur Pharma AG, Grabenstrasse 11A, 8952 Schlieren, Switzerland; paola.atzei@topadur.com (P.A.); reto.naef@topadur.com (R.N.)

**Keywords:** autophagy, cGMP, fibroblasts, mass spectrometry, nitric oxide, NO, proteomics, skin, sildenafil, wound healing

## Abstract

Wound healing pathologies are an increasing problem in ageing societies. Chronic, non-healing wounds, which cause high morbidity and severely reduce the quality of life of affected individuals, are frequently observed in aged individuals and people suffering from diseases affected by the Western lifestyle, such as diabetes. Causal treatments that support proper wound healing are still scarce. Here, we performed expression proteomics to study the effects of the small molecule TOP-N53 on primary human skin fibroblasts and keratinocytes. TOP-N53 is a dual-acting nitric oxide donor and phosphodiesterase-5 inhibitor increasing cGMP levels to support proper wound healing. In contrast to keratinocytes, which did not exhibit global proteome alterations, TOP-N53 had profound effects on the proteome of skin fibroblasts. In fibroblasts, TOP-N53 activated the cytoprotective, lysosomal degradation pathway autophagy and induced the expression of the selective autophagy receptor p62/SQSTM1. Thus, activation of autophagy might in part be responsible for beneficial effects of TOP-N53.

## 1. Introduction

Physiological wound repair is a complex process that aims to restore tissue integrity and function after injury. Skin wound healing is divided into three overlapping phases: inflammation, new tissue formation, and tissue remodeling. In the inflammation phase, neutrophils and macrophages are recruited, which defend damaged tissue against invading pathogens. In addition, they release growth factors and cytokines to promote tissue-regeneration and proliferation in the tissue formation phase. Skin keratinocytes become activated and migrate beneath the fibrin clot to close the epidermal gap. Finally, remodeling and differentiation of the new skin occurs [1]. This allows efficient regeneration of the epidermis, whereas the repair of the dermis lags behind epidermal healing. Proliferation of dermal fibroblasts and migration into the wound is only initiated after re-epithelialization and results in scar formation [2,3,4]. Defects in wound healing and the appearance of chronic wounds is an increasing problem in Western societies, which can be linked to ageing and the growing incidence of age-associated diseases, such as diabetes. Chronic wounds cause high morbidity, severely reduce life quality of affected individuals, and cause enormous costs to healthcare systems [5,6].

Despite being an increasing health problem, treatment of wound healing disorders is often unsatisfactory and limited to symptomatic therapies, mainly because the molecular and cellular processes underlying wound repair remain insufficiently understood [7]. Wound healing is driven, in part, by the coordinated release of a complex mixture of growth factors and cytokines. Also, small diffusible messenger molecules such as nitric oxide (NO) [8] and its downstream signal transducer, the second messenger cyclic guanosine monophosphate (cGMP) [9], have been identified as important in physiological wound repair. Several studies have demonstrated that NO plays an essential role in the regulation of different stages of wound healing, including anti-inflammatory and anti-microbial responses, cell proliferation, re-epithelialization, collagen formation, and angiogenesis. Reduced NO production, as in diabetic patients, has been linked to impaired wound healing and development of chronic wounds [10,11]. Therefore, great efforts are undertaken to increase NO levels for favoring wound healing, e.g., by the development of efficient NO-donors and proper delivery systems [11]. The beneficial role of its downstream signal transducer cGMP in wound healing has also been shown. The topical or systemic application of inhibitors of the cGMP-degrading enzyme phosphodiesterase 5 (PDE5), such as sildenafil (sold as Viagra), enhanced contraction, re-epithelialization, and tensile strength of rat wounds [12], and reduced pressure ulcer size in few patients [13].

Given the promising results shown by NO donors and PDE5 inhibitors in ameliorating wound repair in several model systems, a dual-acting compound has been developed, TOP-N53 [14,15]. The aim of this compound is to preserve high cGMP levels in wounds by both (i) activating the cGMP-producing enzyme soluble guanylate cyclase (sGC) by providing NO and (ii) inhibiting the cGMP-degrading enzyme PDE5 (Figure 1a) [14]. TOP-N53 is marketed by TOPADUR Pharma AG and entered phase 1 clinical trial in September 2020. In October 2021, the European Medicines Agency (EMA) granted TOP-N53, the orphan drug designation for digital ulcers in systemic sclerosis patients. The wound-healing-promoting capacity of TOP-N53 has been demonstrated to be based on direct and indirect effects on the major skin-resident cell types fibroblasts and keratinocytes. It was designed for local administration and its half-maximal inhibitory concentration for PDE5 (IC_50_) was determined to be 1.6 nM, approximately five-fold lower than for the clinically approved PDE5 inhibitor sildenafil [15]. However, the detailed molecular pathways triggered by TOP-N53 in skin cells are still under debate or remain to be elucidated.

In this work we apply unbiased expression proteomics to define potential molecular pathways activated by TOP-N53 in primary human skin fibroblasts and keratinocytes. We provide evidence that in primary skin fibroblasts, TOP-N53 is capable of activating autophagy, a catabolic cytoprotective lysosomal degradation pathway that is implicated in proper wound healing [16]. Thus, TOP-N53 supports wound repair in a multilayered fashion, which might be one reason for the robust responses observed in mice.

## 2. Results

### 2.1. Effects of TOP-N53 on Primary Skin Cells

TOP-N53 is a bifunctional NO-releasing nitrate designed to act as an NO donor and PDE5 inhibitor [14], thereby increasing cGMP synthesis and at the same time preventing its degradation (Figure 1a,b). To determine a suitable, non-toxic concentration of TOP-N53 in our experimental setup, we assayed the viability of primary normal human skin fibroblasts (NHF) and keratinocytes (NHK) isolated from a healthy donor using increasing concentrations of TOP-N53. Cells were treated for 24 h and 48 h, starting with a concentration of 1 nM up to 1 μM. Higher concentrations of TOP-N53 were not considered since this compound is expected to be applied topically to wounds at μM concentrations [15]. None of the tested concentrations appeared to be toxic for the cells. On the contrary, increasing doses of TOP-N53 induced cell proliferation, more markedly in NHF than in NHK (Figure 1c,d). Since we observed a robust pro-proliferative effect in NHF at both 24 h and 48 h with 0.1 μM of TOP-N53, increasing concentrations not leading to additional benefits, we decided to use this concentration for subsequent expression proteomics.

### 2.2. Proteomic Profiling Reveals Differentially Regulated Pathways in TOP-N53-Treated NHF

To test whether skin cells exhibit a differential proteome profile upon TOP-N53 treatment, we performed global, label-free expression proteomics using a standard bottom-up proteomics workflow (Figure 2a). NHF and NHK were treated for 24 h with vehicle (0.1% DMSO) or 0.1 μM TOP-N53, three biological replicates for each experimental group. Using a data-dependent acquisition workflow and label-free quantification (LFQ), we identified in total 4732 protein groups in NHF, of which 3119 were present in all samples and considered for further analyses (Appendix A). A principal component analysis addressing the variability between the individual samples revealed that NHF treated with vehicle (Control) or TOP-N53 were separated into two distinct groups, indicating that TOP-N53 treatment led to global alterations of the cellular proteome (Figure 2b). As this was not the case for NHK (Appendix A), we focused our further analyses on NHF.

An unsupervised hierarchical cluster analysis of protein abundances also split the samples into two distinct groups, treated vs. non-treated NHF (Figure 2c). Gene Ontology (GO) analysis covering biological processes (GOBP) and cellular components (GOCC) revealed a significant enrichment of terms related to vesicle trafficking and lysosomal degradation upon TOP-N53 stimulation, indicating that TOP-N53 might support cytoprotection by activating autophagy (Figure 2c, Appendix A). On the other hand, cells that remained untreated presented an enrichment in GO-terms that might point to a less motile phenotype compared to the TOP-N53-treated NHF (Appendix A), which agrees with published data highlighting increased migration of NHF after TOP-N53 treatment [15].

### 2.3. Fibroblasts Treated with TOP-N53 Exhibit Increased Autophagic Flux

Comparing protein abundances between treated and non-treated NHF, we identified 144 significantly differentially regulated proteins (Figure 3a; *t*-test, FDR < 0.05, permutation-based, Appendix A). Focusing on proteins that were significantly upregulated in TOP-N53-treated NHF, we found proteins related to autophagy, such as the selective autophagy receptors (SARs) SQSTM1/p62 and OPTN and the lysosomal membrane protein LAMP2 [17]. Using these upregulated proteins, we were able to generate a protein interaction network of 63 members, corroborating the hypothesis that TOP-N53 might activate autophagy-related mechanisms (Figure 3b; Appendix A).

An abundance increase in SARs could either indicate increased autophagy induction, i.e., an increase in autophagy flux, or a blockage of autophagy-dependent protein degradation, i.e., a blockage of autophagy flux [18]. To discriminate these two scenarios, NHF were treated with TOP-N53 in the presence or absence of concanamycin A (ConA), an inhibitor of V-ATPase blocking lysosomal activity. Since phenotypical analyses are often less sensitive than the MS-based studies, we used 1 μM of TOP-N53 for NHF stimulation in order to better observe differences in autophagic flux. First, we studied the accumulation of p62 aggregates in NHF by immunofluorescence microscopy. We observed that treatment with TOP-N53 led to an accumulation of p62 puncta (Figure 3c,d), verifying the results obtained by MS-based proteomics (Figure 3a,b). Importantly, treatment with TOP-N53 led to an accumulation of p62 puncta, which was further increased by treatment with ConA, indicating that the TOP-N53 treatment leads to an activation of functional autophagy (Figure 3c,d). This was also observed by monitoring the abundance of the lipidated version of the autophagosome marker protein MAP1LC3B (LC3B-II) using quantitative immunoblotting (Figure 3e,f). To relate the effect of TOP-N53 to known stimuli leading to an induction of functional autophagy, we treated cells with the mTORC1 inhibitor Torin-1 and compared LC3B-II levels [19]. ConA was added in all cases for 2 h. Twenty-four hours of TOP-N53 treatment had a similar effect as 2 h of Torin-1 treatment (Figure 3g,h), indicating that a prolonged treatment with TOP-N53 leads to an activation of functional autophagy in NHF comparable in strength to an acute blockage of mTORC1 signaling. Inhibition of the autophagy-specific kinase complex ULK1/2 with MRT68921 served as a negative control since it impedes autophagy initiation.

The increased abundance of p62 upon treatment with TOP-N53 could be regulated on a transcriptional and/or post-transcriptional level. To address this question, we performed a qPCR analysis and showed that *P*62 mRNA levels were significantly increased upon TOP-N53 stimulation (Figure 4a, *p* < 0.05, *n* = 3), suggesting that the accumulation of p62 protein levels could be due, at least in part, to the activation of gene transcription. Noteworthy, ConA treatment also induced *P*62 transcription (Figure 4a). To address the underlying signaling supporting TOP-N53-driven autophagy, we tested the phosphorylation state of the VPS34 lipid kinase complex member ATG14 at the residue Ser29, which is a ULK1/2 target site [20]. Indeed, we observed an increase in phosphorylation levels of ATG14-Ser 29 in NHF upon TOP-N53 treatment (Figure 4b,c). In contrast, and as expected, blockage of ULK1/2 kinase with the inhibitor MRT68921 prevented the phosphorylation of ATG14 at the residue Ser29 in control conditions as well as in TOP-N53-treated cells (Figure 4d). The ULK1/2 target site Ser318 of its complex member ATG13 behaved similarly (Appendix A), indicating activation of the ULK1/2 kinase complex and autophagy by TOP-N53 treatment. In agreement, inhibition of ULK1/2 also led to a further accumulation of p62 in NHF due to a blockage of autophagy flux, as well as to a block in the increase in LC3B-II levels after TOP-N53 stimulation in NHF.

To test if next to ULK1/2 also the upstream kinase complex mTORC1 was affected by TOP-N53 treatment, we compared its effects to Torin-1 and monitored the mTORC1 target sites pSer757 on ULK1 itself, pThr389 on RPS6KB1, and pThr37/46 on EIF4EBP1 (Appendix A). The two former sites were less phosphorylated in TOP-N53-treated samples; however, compared to Torin-1 treatment, the effects were moderate. Thus, TOP-N53 treatment leads at least in part to the induction of canonical, ULK1-dependent functional autophagy, with minor inhibitory effects on mTORC1 activity.

### 2.4. The Bifunctional Mode-of-Action of TOP-N53 Contributes to Autophagy Induction in Skin Fibroblasts

As stated before, TOP-N53 has a dual effect acting as NO donor and PDE5 inhibitor to increase cGMP levels (Figure 1a). To distinguish the contribution of NO-donor activity from PDE5 inhibition of TOP-N53 on the activation of autophagy, we performed experiments monitoring autophagy in the presence of TOP-N53 and uric acid (UA), a known NO scavenger [21], or sildenafil, a PDE5 inhibitor [12]. First, we used UA to block potential effects provoked by increasing NO levels in NHF due to TOP-N53 treatment and observed that UA was able to partially interfere, although not significantly, with the TOP-N53-dependent increase in *P*62 mRNA levels (Figure 5a). Corroborating the interpretation that NO elevation contributes, at least in part, to increased autophagy, UA also interfered with the accumulation of p62 and LC3B-II protein levels in NHF upon exposure to TOP-N53, as well as with the phosphorylation of ATG14 (Figure 5b). Nonetheless, the blockage of TOP-N53 effects by UA was only partial and not complete (Figure 5a,b), suggesting that additional effects of TOP-N53 contribute to the activation of autophagy.

To test the implication of PDE5 inhibition on autophagy induction driven by TOP-N53, we analyzed the effects in NHF triggered by sildenafil. Sildenafil is a specific PDE5 inhibitor blocking cGMP breakdown and mimicking the effects of TOP-N53 due to the inhibition of PDE5 only. Sildenafil was recently shown to activate autophagy in an experimental mouse model of autoimmune encephalomyelitis [22]. As TOP-N53, sildenafil led to an increase in *P*62 mRNA levels (Figure 5c). In addition, sildenafil also induced an elevation in the protein levels of p62 and LC3B-II in NHF (Figure 5d), which further accumulated when lysosomal degradation was blocked by ConA, pointing to an activation of autophagy. ULK1 activity, as analyzed by the phosphorylation level of ATG14, seemed not to be increased (Figure 5d), suggesting that sildenafil and TOP-N53 have similar, yet not identical, effects on autophagy activity. Thus, both mechanisms of action of TOP-N53 appear to support the activation of functional autophagy in skin fibroblasts.

## 3. Discussion

Despite wound healing pathologies being an increasing problem in developed countries [5,6], therapeutic interventions for treatment of chronic wounds are still unsatisfactory [23], which is at least in part due to the insufficient understanding of the complex underlying mechanisms. In this work, we used proteomics to study the molecular pathways triggered by the small molecule TOP-N53 in primary skin fibroblasts. TOP-N53 possess a potent wound-healing-promoting capacity, inducing an increase in skin cell proliferation, migration, and contraction of the ECM [15]. Whereas it is known that TOP-N53 leads to an increase of the wound-healing-promoting second messenger cGMP by both increasing its synthesis and preventing its degradation [14], affected molecular pathways are less well characterized.

TOP-N53 induces a profound response in the proteome of primary skin fibroblasts, in contrast to keratinocytes. Despite TOP-N53 being shown to increase the proliferative and migratory capacities of keratinocytes [15], we did not observe substantial differences in their proteome, which might be due to a lack of *PDE*5 expression [24,25]. In contrast, in NHF, TOP-N53 stimulation leads to an accumulation of autophagy-relevant proteins and a downregulation of proteins related to cell adhesion. The latter could point to a more motile phenotype of TOP-N53-treated cells as has been observed by others [15]. Proteins important in the consecution of autophagy, autophagosome formation, and degradation, such as OPTN, p62/SQSTM1, or LAMP2 were significantly upregulated. OPTN and p62/SQSTM1 are classical SARs, changes in their abundance being a hallmark of changes in autophagy activity [26]. Increased abundance of LAMP2 indicates an increased abundance of lysosomes, which is often linked to increased lysosomal activity, again being indicative for changes in autophagy activity [27]. Thus, those results led us to hypothesize that TOP-N53 could activate autophagy in dermal fibroblasts as a protective, wound-healing-promoting mechanism.

Autophagy in skin has been reported to play a critical role in controlling skin aging and homeostasis by removing aged organelles and regulating the stress responses of keratinocytes, skin fibroblasts, and melanocytes [28,29]. Defects in autophagic flux have been reported in fibroblasts from patients suffering from premature aging and age-related diseases, such as Cockayne syndrome [30]. Moreover, loss of proteostasis has been observed in skin fibroblasts upon chronologic and UV-related exposure. Autophagy has also been reported to regulate ECM metabolism in skin fibroblasts, which can also influence wound healing [31,32]. Indeed, the role of autophagy in regulating the different phases of wound healing is increasingly recognized, although deeper understanding of the molecular mechanisms is necessary to promote a fine-tuned regulation in the different cells and wound-healing stages. Only then would a therapeutic intervention in pathological situations such as refractory wounds be conceivable [16].

To corroborate that TOP-N53 is inducing autophagy in primary skin fibroblasts, we analyzed the accumulation of the autophagic markers p62 and LC3B-II in the presence or absence of the lysosomal inhibitor ConA. Using immunofluorescence microscopy and Western blots, we verified that TOP-N53 increases p62 and LC3B-II levels in comparison to non-treated cells, and further levels of accumulation were observed in the presence of ConA, i.e., TOP-N53 increased autophagic flux. In agreement, autophagy signaling was increased as monitored by an increase in phosphorylation of ULK1 target sites on ATG13 and ATG14 and a decrease in mTORC1 kinase activities [20,33]. Thus, we corroborated that TOP-N53 induces functional autophagy in a canonical, ULK1-dependent manner.

To understand the increase in p62 protein levels due to TOP-N53 treatment, we analyzed the transcriptional regulation of *P*62 and showed that TOP-N53 increased its mRNA levels. One potential transcription factor that might be involved in this regulation is Nrf2, a major regulator of cytoprotective responses caused by reactive oxygen species (ROS) and electrophiles [34]. The promoter of *P*62 contains AREs (Antioxidant Response Elements) that can be targeted by Nrf2 to induce its transcription [35,36]. Also, the E3 ligase Keap1, a negative regulator of Nrf2, has been shown to be degraded by p62-dependent selective autophagy [35]. This would create a positive feedback that connects oxidative stress response and autophagy in TOP-N53-treated cells [35,36,37]. Whether TOP-N53 promotes the Nrf2-dependent upregulation of p62 will have to be addressed in future studies.

As TOP-N53 is a bifunctional compound acting as an NO donor and PDE5 inhibitor, we assessed whether one or both biochemical actions were contributing to the observed effects of TOP-N53 on autophagy. By using the NO scavenger UA [21], we found that elevation of NO levels in NHF upon TOP-N53 appear in part responsible for autophagy activation. Moreover, when we treated NHF with the potent PDE5-specific inhibitor sildenafil, we corroborated that PDE5 inhibition induced a similar response to TOP-N53, resulting in autophagy activation [22]. Thus, both TOP-N53-induced mechanisms appear to contribute to an activation of autophagy, which is likely due to an increase in *P*62 transcription. How cGMP elevation induces this response is a question we will address in the future.

## 4. Material and Methods

### 4.1. Primary Skin Cells Culture

Normal human primary fibroblasts (NHF) and keratinocytes (NHK) were isolated from foreskin. Fibroblasts were cultured in DMEM with 1 g/L glucose, L-glutamine and sodium pyruvate (PAN-Biotech, Aidenbach, Germany) supplemented with 10% fetal bovine serum (FBS) (ThermoFisher Scientific, Waltham, MA, USA) and 1% Penicillin/Streptomycin (PAN-Biotech, Aidenbach, Germany). Keratinocytes were cultured in Keratinocyte Serum-Free Medium (SFM 1× Gibco; ThermoFisher Scientific, Waltham, MA, USA) supplemented with human recombinant epidermal growth factor (rEGF) and bovine pituitary extract (BPE) and 1% Penicillin/Streptomycin. Cells were maintained at 37 °C in a 5% CO_2_-humidified atmosphere and harvested with trypsin/EDTA solution (0.05%/0.02% weight/volume) (PAN-Biotech, Aidenbach, Germany).

### 4.2. Reagents

TOP-N53 was synthesized as described previously [14]. TOP-N53 was diluted to 10 mM in sterile dimethylsulfoxide (DMSO; Sigma-Aldrich, St. Louis, MO, USA), and further dilutions in DMSO or cell culture media were done to reach the desired concentrations for treatments. The final DMSO concentration was 0.1%. Concanamycin A (ConA) was purchased from Santa Cruz Biotechnologies (Heidelberg, Germany) and resuspended in DMSO. MRT68921 (ULK1 inhibitor) was purchased from Selleckchem (Houston, TX, USA) and resuspended in DMSO. Uric acid (UA) and Sildenafil were purchased from MedChemExpress (Sollentuna, Sweden), the latter one being dissolved in DMSO and UA following manufacturer’s instructions. Torin-1 was purchased from Tocris (Bristol, UK) and resuspended in DMSO. Primary antibodies anti-phospho-ATG14 (Ser29) (D4B8M; #92340), anti-SQSTM1/p62 (#5114), anti-phospho-p70 S6 Kinase/RPS6KB1 (Thr389) (1A5; #9206), anti-phospho-ULK1 (Ser757) (#6888), anti-phospho-4E-BP1 (Thr37/46) (236B4; #2855), and anti-4E-BP1 (#9452), were purchased from Cell Signaling Technology (St. Louis, MO, USA). Anti-LC3 (5F10) antibody was purchased from NanoTools (Teningen, Germany), anti-phospho-ATG13 (Ser318) (600-401-C49) was purchased from Rockland Immunochemicals (Pottstown, PA, USA), and anti-GAPDH (FL-335; sc-25778) was purchased from Santa Cruz Biotechnologies (Heidelberg, Germany). For immunostaining, anti-p62/SQSTM1 (C-terminus) guinea pig polyclonal (GP62-C) was purchased from Progen (Heidelberg, Germany). Horseradish peroxidase (HRP)-conjugate secondary antibodies anti-rabbit (111-035-045) and anti-mouse (115-035-062) were purchased from Jackson ImmunoResearch Laboratories (Newmarket, UK). Alexa Fluor 488 anti-guinea pig (A11073) was purchased from Invitrogen Life Technologies.

### 4.3. Cell Viability Assay

NHF and NHK were cultured in 96-well plates and NHF were serum deprived (0.25% FBS) prior to the treatment. Cells were treated with increasing concentrations of TOP-N53 for 24 h and 48 h. When 48 h of treatment was performed, the medium containing TOP-N53 was refreshed after 24 h. After the incubation time, viability was measured with Cell Counting Kit 8 (CCK8) (Dojindo EU GmbH, Munich, Germany) following the manufacturer’s instructions.

### 4.4. MS Sample Preparation

NHF and NHK were cultured in 10 cm^2^ plates and left to grow until they reached ~70% confluence. NHF were serum starved (0.25% FBS) prior to the treatment with vehicle (0.1% DMSO) or 0.1 μM TOP-N53 for 24 h. Cells were lysed with 1% Na-deoxycholate prepared in 50 mM ABC buffer (50 mM Ammonium bicarbonate, pH7.5). 1 μL of benzonase was added to the cell lysates to digest nucleic acids. Protein concentration was determined by pierce BCA protein assay kit and samples containing 75 μg protein were prepared. Samples were reduced with 1 mM DTT and alkylated with 5.5 mM iodoacetamide (Sigma-Aldrich, St. Louis, MO, USA), and they were digested with trypsin HPLC-grade (Promega AG, Dübendorf, Switzerland) overnight. Next, Na-deoxycholate was precipitated with 1% v/v of formic acid and eliminated by 10 min centrifugation at a max. speed of 13,000 rpm. The resulting peptide solutions (supernatants) were desalted on C18-based STAGE tips as previously described [38] and analyzed by liquid chromatography-MS/MS.

### 4.5. Mass Spectrometry

LC-MS/MS measurements were performed on a QExactive HF-X mass spectrometer coupled to an EasyLC 1200 nanoflow-HPLC. Peptides were separated on a fused silica HPLC-column tip (I.D. 75 μm, New Objective, self-packed with reprosil-Pur 120 C18-AQ, 1.9 μm (Dr. Maisch) to a length of 20 cm) using a gradient of A (0.1% formic acid in water) and B (0.1% formic acid in 80% acetonitrile in water). Peptides were separated by 5–30% B within 85 min with a flow rate of 250 nL/min. Spray voltage was set to 2.3 kV and the ion-transfer tube temperature to 250 °C; no sheath and auxiliary gas were used. The mass spectrometer was operated in the data-dependent mode; after each MS scan (mass range *m*/*z* = 370–1750; resolution: 120,000), a maximum of 12 MS/MS scans were performed using a normalized collision energy of 25%, a target value of 5000, and a resolution of 30,000.

### 4.6. Identification of Proteins Using MaxQuant

Peak detection and relative quantification of peptides and proteins was conducted by MaxQuant version 1.6.2.10 (Max Planck Institute of Biochemistry, Planegg, Germany) [39]. Peaks were searched against a full-length human Uniprot database from March 2016. Enzyme specificity was assigned to trypsin/P and quantitation was based on Label Free Quantification (LFQ). Carbamidomethylcysteine was set as fixed modification, and protein amino-terminal acetylation and oxidation of methionine were set as variable modifications. The MS/MS tolerance was set to 20 ppm. A maximum of three missed cleavages was allowed. Proteins were identified with a minimum of one unique peptide. For protein quantification, a ratio count of two was set. Identification and quantification of peptides and proteins was based on a reverse-database with a false discovery rate (FDR) of 0.01. Peptide length had to be at least seven amino acids. Identified peptides were re-quantified. For enhanced identification, runs were matched with a match time window of 1 min.

### 4.7. Proteomics Data Analysis

Obtained data was analyzed using Perseus (Max Planck Institute of Biochemistry, Planegg, Germany) 1.6.6.0 [40]. Contaminants, reverse hits, and proteins only identified by site were excluded from further analysis. Data were log2 FDR of 0.05. For cluster analysis, protein intensities were z-normalized. Rows and column trees were clustered using Euclidean distance, with no constraints and average as linkage. The number of clusters was set to 300 and the maximum number of iterations to 10. Protein clusters were set using the distance threshold to capture the major differences between the two experimental groups. Proteins in each hierarchical cluster were tested for enriched Gene Ontology terms using the human proteome as background (biological process and cellular compartment, minimum protein count of 5). Enriched GO-terms (FDR < 0.05) were identified in STRING version 11 [41].

### 4.8. Western Blot Analysis

NHF were cultured in 6 cm^2^ plates or 6-well plates until they reached ~70% confluence. Cells were serum starved (0.25% FBS) prior to stimulation with TOP-N53 or sildenafil, ConA, MRT68921, Torin-1, and uric acid alone or in combination as indicated in the results section and figure legends. If not stated otherwise, the used default concentrations were: TOP-N53, 1 μM; UA, 40 mg/L; ConA, 2.5 nM; Sildenafil, 100 μM), MRT68921, 1 μM; and Torin-1, 1 μM. Cells were lysed with modified RIPA buffer (50 mM Tris-HCl pH 7.5, 150 mM NaCl, 1 mM EDTA, 1% NP-40, 0.25% deoxycholic acid; Sigma-Aldrich, St. Louis, MO, USA) supplemented with proteases and phosphatases inhibitors. Lysates were centrifuged at 12,000 rpm for 15 min at 4°C. Protein concentration was determined by pierce BCA protein assay kit and, when possible, samples containing 25 μg protein were prepared in Laemmli loading buffer containing 1 mM DTT. Samples were heated for 10 min at 75 °C. Proteins were separated in SDS-PAGE gels and transferred to 0.2 mm PVDF membranes (Amersham™ Hybond™ Protran, Merck KGaA, Darmstadt, Germany ). The membranes were blocked for 1 h in 5% BSA or 5% non-fat milk prepared in TBS with 0.1% Tween-20 (TBST) and they were incubated overnight at 4 °C with the following primary antibodies and dilutions: anti-phospho-ATG14 (Ser29) (1:1000), anti-SQSTM1/p62 (1:1000), anti-LC3 (1:1000), anti-phospho-ATG13 (Ser310) (1:1000), anti-phospho-ULK1 (Ser757) (1:1000), anti-Phospho-p70 S6 Kinase/RPS6KB1 (Thr389) (1:1000), anti-phospho-4E-BP1 (Thr37/46) (1:1000), anti-4E-BP1 (1:1000), and anti-GAPDH (1:1000) as loading control. The next day, blots were incubated for 1 h at room temperature with goat anti-rabbit (1:10,000) or anti-mouse (1:5000) HRP-conjugated secondary antibodies (Jackson ImmunoResearch Europe Ltd., Newmarket, UK) and signal was detected with WesternBright ECL Spray (Advansta, Menlo Park, CA, USA) or SuperSignal West Femto Maximum Sensitivity Substrate (ThermoFisher Scientific, Waltham, MA, USA) in Odyssey^®^ Fc Imaging System (LI-COR Biosciences, Lincoln, NE, USA). Densitometric analysis for quantification was done with Image-J (NIH) software.

### 4.9. Confocal Microscopy

Cells were cultured on collagen I pre-coated coverslips disposed on 24-well plates. In total, 15,000 NHFs per condition were cultured and serum-starved prior to stimulation with 0.1 μM TOP-N53 for 24 h and/or ConA (10 nM) for 2 h. After the treatment, cells were fixed with cold methanol for 15 min and washed with PBS. Blocking and permeabilization was performed for 30 min RT with 5% horse serum prepared in PBS with 0.1% Tween 20 (Sigma-Aldrich, St. Louis, MO, USA). Cells were incubated with anti-p62/SQSTM1 (C-terminus) (1:400) overnight at 4 °C in a humid chamber. Coverslips were washed with PBS with 0.1% Tween 20 and followed by incubation with secondary antibody Alexa Fluor 488 anti-guinea pig (1:2000) together with Hoechst 33342 1 μM (14533; Sigma-Aldrich, St. Louis, MO, USA) for nuclei staining for 2 h at room temperature. Coverslips were mounted on glass slides with ProLong Gold anti-fade reagent (P36930; ThermoFisher Scientific, Waltham, MA, USA), and the edges were sealed with nail polish. Cells were visualized by inverted microscopy. Images of the specimens were collected with a STELLARIS 8 FALCON confocal microscope (Leica Microsystems, Wetzlar, Germany), equipped with equipped with 4 HyD detectors and 63×/1.3 glycerol-immersion objective. Z-series images were obtained through the collection of serial confocal sections at 0.34 μm intervals. To compare the confocal data, identical confocal settings were used for image acquisition of different experiments. Quantitation of p62 dots was performed with Imaris ×64 v7.2.1 software (BITPLANE, Oxford Instruments). For the generation of p62 spots, particle size was measured and determined to be 0.75 μm, and the number of spots were counted per cell.

### 4.10. mRNA Isolation and Reverse Transcriptase qRT-PCR

NHF were treated with vehicle, 1 μM of TOP-N53, 100 μM of Sildenafil, and/or 40 mg/L of uric acid for 24 h and supplemented with 2.5 nM of ConA (2 h) when indicated. RNA was isolated with RNeasy Mini kit (Qiagen, Hilden, Germany) following the manufacturer’s guidelines. Reverse transcription was performed with Quantitect RT kit (Qiagen, Hilden, Germany), using 400 ng RNA as starting material. Quantitative PCR was then performed using the KAPA SYBR Fast (Universal) qPCR kit (Merck Millipore, Burlington, MA, USA) according to the manufacturer’s recommendations. The qRT PCR reaction was performed with a Rotor-GeneQ (Qiagen, Hilden, Germany) and the same thermal profile conditions were used for all primer sets: 95 °C for 10 min; then 40 cycles were performed of 10 s at 95 °C, 20 s 60 °C, and 20 s 72 °C. The following primer pairs were used: *HPRT*1 forward 5′-TGA CAC TGG CAA AAC AAT GCA-3′; *HPRT*1 reverse 5′-GGT CCT TTT CAC CAG CAA GCT-3′; *P*62 forward 5′-CAT CGG AGG ATC CGA GTG TG-3′; and *P*62 reverse 5′-TTC TTT TCC CTC CGT GCT CC-3′.

### 4.11. Statistical Analysis

Graph-Pad Prism 6 software (GraphPad, La Jolla, CA, USA) was used for graphs generation and statistical analysis. Data are presented as mean ± s.e. (standard error) of the number of indicated experiments, unless otherwise mentioned. Appropriate statistical tests were performed based on the compared experimental conditions, including Student *t*-test (computed with FDR ≤ 0.05 for correcting for multiple testing when necessary) or one-way analysis of variance (ANOVA). Throughout the paper, statistically significant differences are indicated by: * *p* ≤ 0.05 and ** *p* ≤ 0.01.

## 5. Conclusions

In the current study, we analyzed the effects of TOP-N53 on keratinocytes and skin fibroblasts in 2D cell culture using expression proteomics. Whereas keratinocytes did not respond by global alterations of their proteome to TOP-N53 treatment, fibroblasts exhibited an increase of autophagy- and lysosome-associated proteins. Both its mechanisms of action, an increase in NO levels and inhibition of cGMP degradation, appear to contribute to its activatory effects on autophagy in skin fibroblasts. How this affects wound healing and if autophagy plays a role in fibroblast-keratinocyte communication will be addressed in future studies.

As it has been shown that 3D cell culture better recapitulates in-vivo observations and as 3D cell culture has a profound effect on the composition of the proteome [42,43], additional new insights can be generated by the use of 3D cell culture systems. To better understand potential mechanisms of action, separate 3D cell cultures might still be the method of choice. However, as both keratinocytes and fibroblasts closely interact in vivo and as their interaction is vital for skin biology, it will be important to study the drug effects in organotypic co-culture systems [44], not only by expression proteomics but also by phosphoproteomics, to map underlying signal transduction events being causal for observed phenotypes.

## Figures and Tables

**Figure 1 ijms-23-06860-f001:**
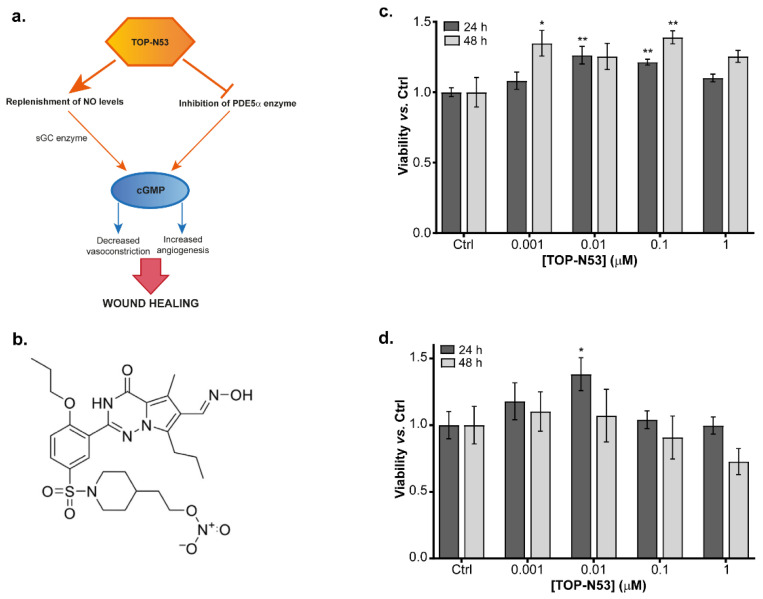
Effect of TOP-N53 in primary skin cells. (**a**) Dual mechanism of action of TOP-N53 acting as NO-donor and PDE5 inhibitor increasing cGMP levels to promote wound healing. (**b**) Chemical formula of TOP-N53. (**c**) Viability assay of primary NHF and (**d**) NHK with increasing concentrations of TOP-N53 at 24 h and 48 h. Data represented as mean ± s.e. (*n* = 3). * *p* ≤ 0.05 and ** *p* ≤ 0.01 using one-way ANOVA with a Dunnet test post-hoc analysis.

**Figure 2 ijms-23-06860-f002:**
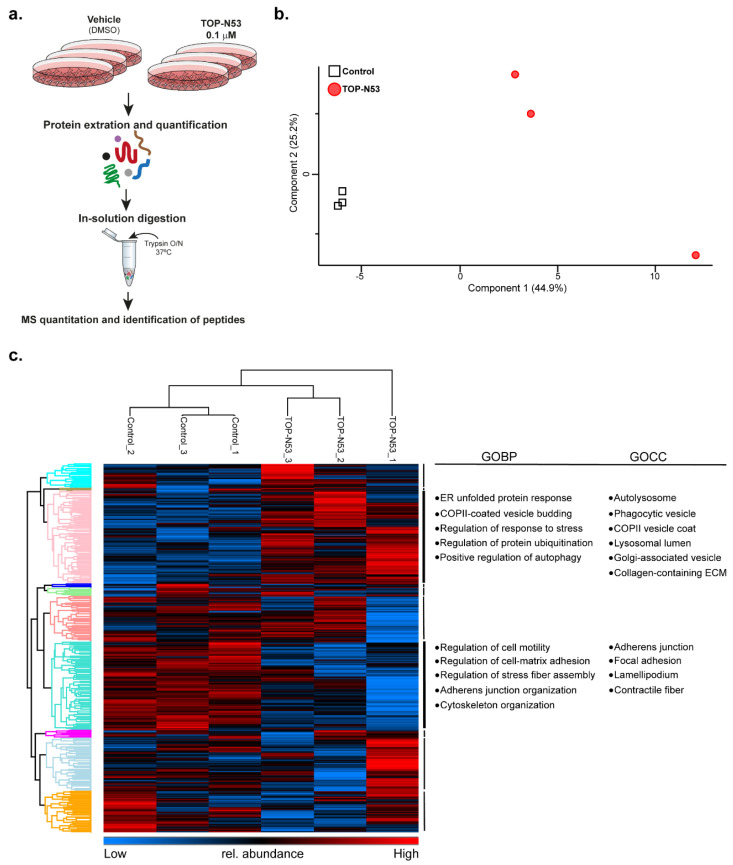
Expression proteomics reveals differentially regulated pathways in TOP-N53 treated NHFs. (**a**) Label-free MS-based workflow of the analysis of the effects of TOP-N53 treatment. Three replicates of control (DMSO) and 0.1 μM TOP-N53-treated (24 h) NHF were used for the analysis. Samples were processed as outlined. (**b**) PCA of NHF samples. Log_2_-transformed LFQ intensities of proteins were used as input. The two experimental groups are clearly separated. (**c**) Hierarchical clustering clearly separates control from TOP-N53-treated NHFs. Protein intensities were log_2_ transformed and Z-score normalized. Columns and rows were hierarchically clustered. Proteins in the annotated clusters (color coded in the row dendrogram) were tested for enrichment of GOCC and GOBP terms. Full list of terms for selected clusters is shown in Appendix A.

**Figure 3 ijms-23-06860-f003:**
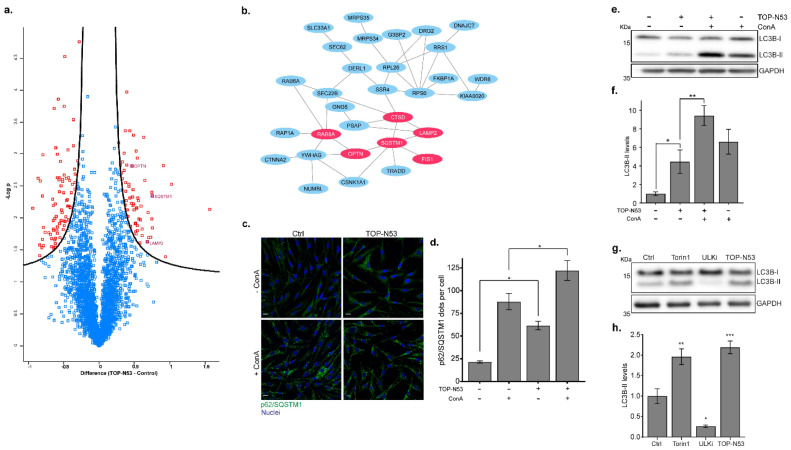
NHFs treated with TOP-N53 exhibit increased autophagic flux. (**a**) Volcano plot highlighting significantly regulated proteins. Data were log_2_ transformed and significance was tested with two-samples test using a permutation-based FDR ≤ 0.05. Proteins enriched in TOP-N53-treated NHFs are placed on the right side and proteins enriched in control conditions are placed on the left side. Proteins related to autophagy (enriched in TOP-N53 treated NHF), such as SQSTM1/p62, OPTN or LAMP2, are highlighted. Full lists of significantly regulated proteins are shown in Appendix A. (**b**) Network analysis of significantly enriched proteins in TOP-N53-treated NHFs. Protein–protein interactions of 63 significantly enriched proteins in TOP-N53-treated NHF as analyzed by STRING DB (FDR < 0.05). Red-labeled proteins carry terms “autophagy” (GOBP). Only part of the network is presented; the list of proteins and full network is shown in Appendix A, respectively. (**c**) Immunofluorescence staining of p62. Control and TOP-N53-treated NHFs in presence or not of ConA were stained with anti-p62 antibody (green). Nuclei were stained with Hoechst 33258 (blue). Scale bars, 15 μm. (**d**) Quantification of p62 puncta. Quantification was performed with Imaris software. Bars represent the mean ± s.e. of the number of p62 puncta per cell in each condition. Between 97 and 127 cells were counted per condition. * *p* ≤ 0.05 using one-way ANOVA with a Tukey test post-hoc analysis. (**e**) Autophagy flux analysis. To analyze lysosomal activity, autophagy flux was analyzed in control and TOP-N53-treated (1 μM; 24 h) NHF in presence or not of 2.5 nM ConA. Representative Western blot image and (**f**) quantification of LC3B-II levels are shown. Data were normalized to GAPDH and are represented as mean ± s.e. (*n* = 3). * *p* ≤ 0.05 and ** *p* ≤ 0.01 using one-way ANOVA with a Holm-Sidak test post-hoc analysis. (**g**) Autophagy activation by TOP-N53 was compared with the induction by the mTORC1 inhibitor Torin-1. NHF were treated with TOP-N53 (1 μM; 24 h), ULK1 inhibitor MRT68921 (1 μM; 2 h), and Torin-1 (1 μM; 2 h). ConA was added in all cases for the last 2 h of treatment. Representative Western blot images and (**h**) quantification of LC3B-II levels are shown. Data were normalized to GAPDH and are represented as mean ± s.e. (*n* = 3). * *p* ≤ 0.05 and ** *p* ≤ 0.01, *** *p* ≤ 0.001 using one-way ANOVA with a Holm–Sidak test post-hoc analysis.

**Figure 4 ijms-23-06860-f004:**
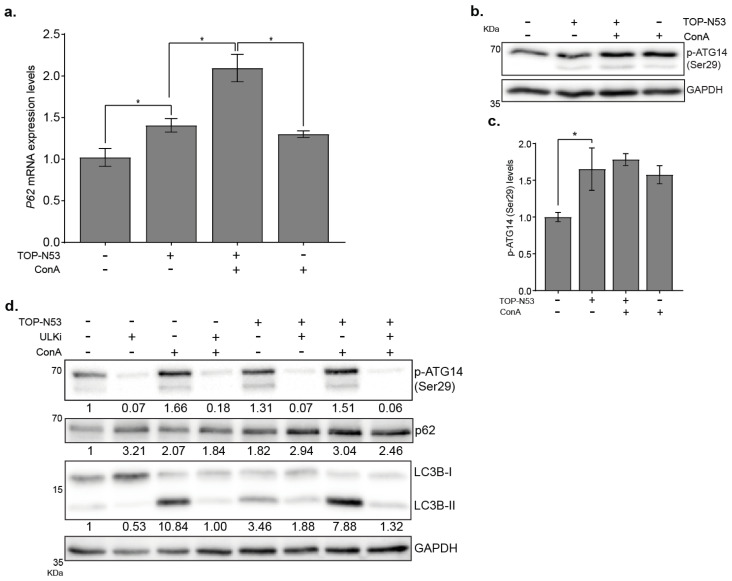
TOP-N53 activates canonical autophagy in NHFs. (**a**) *P*62 mRNA levels in NHFs upon TOP-N53 treatment (1 μM, 6 h) in presence or absence of ConA (2.5 nM, 2 h). Data were normalized to *HPRT*1 levels and are represented as mean ± s.e. (*n* = 3). * *p* ≤ 0.05 using one-way ANOVA with a Holm–Sidak test post-hoc analysis. (**b**) Analysis of ATG14 Ser29 phosphorylation in control and TOP-N53-treated (1 μM; 24 h) NHF in presence or not of 2.5 nM ConA. Representative Western blot image and (**c**) quantification of p- ATG14 levels. Data were normalized to GAPDH and are represented as mean ± s.e. (*n* = 3). * *p* ≤ 0.05 using one-way ANOVA with a Holm–Sidak test post-hoc analysis. (**d**) Analysis of the autophagic flux in NHFs upon treatment with TOP-N53 (1 μM, 24 h) in presence or absence of ConA (2.5 nM, 2 h) and blockade of ULK1 with the specific inhibitor MRT68921 (1 μM, 2 h). Representative Western blot (*n* = 2) and its quantification below each protein panel (normalized to GAPDH).

**Figure 5 ijms-23-06860-f005:**
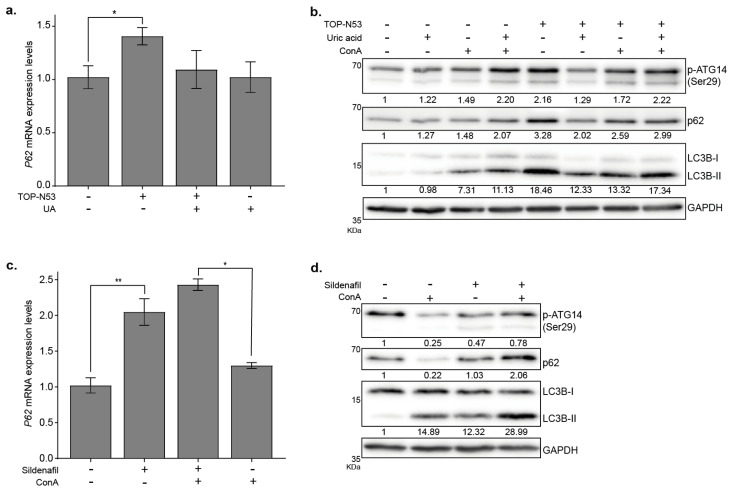
TOP-N53 induces autophagy in NHFs by its dual mechanism of action. (**a**) *P*62 mRNA levels in NHF upon TOP-N53 treatment (1 μM, 6 h) in presence or absence of uric acid (UA, 40 mg/L, 6 h). Data were normalized to *HPRT*1 levels and are represented as mean ± s.e. (*n* = 3). * *p* ≤ 0.05 using one-way ANOVA with a Holm–Sidak test post-hoc analysis. (**b**) Analysis of the autophagic flux in NHFs upon treatment with TOP-N53 (1 μM, 24 h) in presence or absence of UA (40 mg/L, 24 h) and ConA (2.5 nM, 2 h). Representative Western blot (*n* = 2) and its densitometric quantification below each protein panel (normalized to GAPDH). (**c**) *P*62 mRNA levels in NHF upon Sildenafil treatment (100 μM, 6 h) in presence or absence of ConA (2.5 nM, 2 h). Data were normalized to *HPRT*1 levels and are represented as mean ± s.e. (*n* = 3). * *p* ≤ 0.05 and ** *p* ≤ 0.01 using one-way ANOVA with a Holm–Sidak test post-hoc analysis. (**d**) Analysis of the autophagic flux in NHF upon treatment with Sildenafil (100 μM, 24 h) in presence or absence of ConA (2.5 nM; 2 h). Representative Western blot (*n* = 2) and its densitometric quantification below each protein panel (normalized to GAPDH).

## Data Availability

The mass spectrometry proteomics data are publicly available and were deposited to the ProteomeXchange Consortium via the PRIDE partner repository under the identifier PXD032893 [45].

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
