# Peer review of "A Dual-Acting Nitric Oxide Donor and Phosphodiesterase 5 Inhibitor Activates Autophagy in Primary Skin Fibroblasts"

_ijms, 2022, doi:10.3390/ijms23126860_

Round 1

Reviewer 1 Report

Wound healing is very important clinical aspect that need to be addressed as there are no effective drugs that can promote wound healing. Authors in the current manuscript titled "A Dual-Acting Nitric Oxide Donor and Phosphodiesterase 5 inhibitor Activates Autophagy in Primary Skin Fibroblasts" is a very important manuscript for the current clinical requirement. Authors have demonstrated TOP-N53, a bifunctional compound acting as NO donor and PDE5 inhibitor, and how both biochemical actions were contributing to the observed effects of TOP-N53 on autophagy and extrapolating the observations to wound healing. The Manuscript is clear and well written however I have few questions that needs to be addressed before this manuscript can be accepted.

  1. Once of the most important aspect of the wound healing is contraction and NO is known to play a key role in contraction. I would recommend authors to validate the effect of TOP-N53 on Collagen Gel Contraction Assay.
  2. ULK1 directly phosphorylates Beclin-1 at Ser 14 and activates the pro-autophagy class III phosphoinositide 3-kinase (PI (3)K), VPS34 complex, to promote autophagy induction and maturation. [PMID 23685627]. It is very important to identify that this is not the dominant pathway. I would recommend authors to check for Beclin-1 phosphorylation. 
  3. ULK1 can auto phosphorylate itself as well as ATG13, validating ULK1 phosphorylation as well atg13 phosphorylation, I would suggest authors to validate them which will improve the understanding of the autophagic events upon TOP-N53 treatment. 
  4. I have another important validation, if authors can use a positive control of autophagy inducing such as Rapamycin or Lithium chloride this would be a determination for the amount of autophagy induction in the cells treated with TOP-N53. I would specifically recommend for the data in the figure 4 and Figure 5.
  5. Minor comment is ULK1 phosphorylates ATG14 at serine 29, this is labelled in manuscript if authors can label in the Figure, it will enhance the readers understanding.

Author Response

We thank the reviewer for her/his interest in our paper and the critical feedback which we address in detail in the attached point-by-point response.

Reviewer 2 Report

Abstract 

The phrase increased incidences in chronic seems loosely connected

From the full 10 lines of the abstract, it presents 4.5 as general information. Please add your results to it instead of loosely explaining it.

Introduction

Line 34 - defects in wound healing... can be linked to longer life expectancy? Phrase meaning should be corrected.

If abnormal NO production is linked to impaired wound healing, why great efforts are invested in development of efficient NO-donors? If the production is natural from the patient, there should be something to impair this natural behaviour. Phrase seems to be incorrect.

The intent of the work in the introduction should be better explained and it is not easily perceived. For example NO is overexpressed in diabetic patients and the authors wants to maintain in safer levels by using in conjunction with a PDE5 blocker. I suggest the authors to rewrite the lines 39-56. It is not clear why cGMP-degrading enzyme PDE5 should be inhibited. 

Also, it would be best for the authors to have a few words on the TOP-N53 compound, as it is a patent product as listed in references. Please describe why it is relevant, and if it is possible to market it, modify, scale up and competitiveness towards Sildenafil.

Results

Missing information on PCA and HCA (materials and methods). In fact, I do not understand how the HCA was performed in figure 2. How did the authors added information on autolysosome for example, or cytoskeleton organization, or even contractile fibre. All of these needs to be on materials and methods. How the authors know that specific cluster is associated to such terms like contractile fibre. The same could be said to section 2.3

Figure 4 should be better explained. There is overexpression of p62 even without TOP-N53, so the authors suggest that the mode of action is from gene transcription. I fail to observe these from the images of Figure 4. Please clearly indicate how is that possible since TOP-N53 pure shows weakly signs from all expressions of LC3B and p-Atg14 on Figure 4.d.

Figure 5, the author states that UA was able to interfere with TOP-N53 but it was not statistically significant. Please be careful with these affirmations. Why signals from UA are so low and not intense? I guess the concentration used was not effective in order to provide affirmation to this element, and it is probably the reason for such values on Fig.5.b

Discussion

The first and second paragraph is a repetition of the introduction. I would focus it more in the authors work instead of introducing the subject yet again in the beginning of this section. 

Why not add the fourth paragraph in the introduction ? Seems loosely connected as well, the authors are yet again introducing the subject.

Where is the information of section 2.3? It just briefly states that based on the results, it is supposed to be autophagic. Due to the screening of proteins, this must be expanded and defend the argument of autophagic action.

materials and methods

sinadefil drug dosage should be provided.

Where is the conclusion section?

Author Response

Thanks for your interest and comments. Please see the attached file for a detailed point-by-point response.

Reviewer 3 Report

I think that the topic is of interest and worth to be published; I have one major isssue: the limitations of this stuyd are not mentioned and discussed in a profound way. Please do so!

What are the requested next steps? A more complex model?...

Please state and discuss, too.

Author Response

We thank the reviewer for her/his feedback and interest in our manuscript. We added a paragraph to the discussion on page 13 of the revised manuscript stating the limitations of our study and outlining important next steps and future directions: to fully understand the effects of TOP-N53 on wound healing more sophisticated model systems have to be used and we plan to employ (a) 3D cell culture systems using single cell types and (b) organotypic skin models using keratinocyte and fibroblast co-cultures (see discussion for more details).

Round 2

Reviewer 1 Report

The authors have addressed all my concerns. 

Reviewer 2 Report

Thank you for modifying the manuscript.

Reviewer 3 Report

No further requests